# Electrical synapses convey orientation selectivity in the mouse retina

Amurta Nath[1] & Gregory W. Schwartz[1,2,3,4]

Sensory neurons downstream of primary receptors are selective for specific stimulus features, and they derive their selectivity both from excitatory and inhibitory synaptic inputs from other neurons and from their own intrinsic properties. Electrical synapses, formed by gap junctions, modulate sensory circuits. Retinal ganglion cells (RGCs) are diverse feature detectors carrying visual information to the brain, and receive excitatory input from bipolar cells and inhibitory input from amacrine cells (ACs). Here we describe a RGC that relies on gap junctions, rather than chemical synapses, to convey its selectivity for the orientation of a visual stimulus. This represents both a new functional role of electrical synapses as the primary drivers of feature selectivity and a new circuit mechanism for orientation selectivity in the retina.

[1] Interdepartmental Neuroscience Program, Northwestern University, Chicago, IL 60611, USA. [2] Department of Ophthalmology, Northwestern University, Chicago, IL 60611, USA. [3] Department of Physiology, Feinberg School of Medicine, Northwestern University, Chicago, IL 60611, USA. [4] Department of Neurobiology, Weinberg College of Arts and Sciences, Northwestern University, Evanston, IL 60208, USA. Correspondence and requests for materials should be addressed to G.W.S. (email: greg.schwartz@northwestern.edu)

Neural circuits rely on both chemical and electrical synapses for inter-neuronal communication[1]. Electrical synapses are most commonly present in dendrites and often aid in lateral signal spread and synchrony among functionally similar cells[2–6]. Gap junctions have many known functions in the central nervous system, including increasing sensitivity of sensory systems[7], participating in escape behaviours[8], generating persistent firing[9], contributing to expansion of receptive fields[10], compensating for sublinear dendritic integration[6], and regulating interneuron excitability[11].

Gap junctions are particularly prevalent in the retina, where all five major cell classes are connected by electrical synapses[12]. In vertebrates, retinal ganglion cells (RGCs) form homologous gap junction networks with other RGCs of the same type and/or heterologous gap junction networks with amacrine cells[13–15]. As in other parts of the central nervous system, both homologous and heterologous electrical networks have been implicated in increasing spike timing synchrony among RGCs[16–22]. Other reported functional roles of RGC gap junctions involve modulation or support of computations that are principally carried by chemical synapses. The directional tuning of ON direction selective (DS) RGCs is strengthened via coupling to polyaxonal amacrine cells[23]. One population of ON-OFF DS RGCs combines chemical and electrical synaptic input to compute lag normalization[24,25]. OFF transient alpha RGCs receive electrical synaptic inputs from four distinct populations of amacrine cells[26], contributing to their transient response following the offset of dim flashes[27]. Heterologous gap junctions have also been implicated in crossover excitation, in which OFF RGCs receive ON input via electrical synapses[28].

Two of the best-studied computations in RGCs are their selectivity for the direction of movement or the orientation of a visual stimulus. Direction selectivity relies principally on inhibition from starburst amacrine cells[29–32]. Orientation selectivity (OS) has been reported to rely on both excitatory and inhibitory synaptic input[33–36]. Here we identify two types of OFF OS RGCs in the mouse retina, and we show that electrical synapses carry OS information to the RGC. Our anatomical and physiological experiments suggest a morphological substrate of this OS signal in a coupled amacrine cell. These results uncover a surprising new function of electrical synapses in conveying feature selectivity to a sensory neuron, and they establish an amacrine cell-RGC circuit critical to the computation of OS.

## Results

**Functional characterization of OFF OS RGCs.** OS RGCs were first described in rabbit retina where they comprise both ON types, responsive to light increments and OFF types, responsive to decrements[34,35,37]. ON OS RGCs were recently identified in mouse[36], so we began by searching for the OFF OS RGCs in this species. During a large-scale survey of the responses of mouse RGCs, we found OFF RGCs with a distinct response to a 200 μm diameter circular spot of light presented from darkness at the receptive field centre for 1 s. (Fig. 1a, left). These OFF cells had a low baseline firing rate and were completely suppressed at light onset. Following light offset, the firing rate gradually reached baseline level and in some cases overshot the baseline rate. This type of response was distinct from the two well-known OFF alpha RGCs (Fig. 1a). For all OFF RGCs recorded, we measured the average baseline firing rate and the peak firing rate at light offset (Methods). These low baseline and low peak firing rate RGCs formed a distinct cluster in the two-dimensional response space and were formally classified by an average baseline firing rate <50 Hz and a peak light offset firing rate <120 Hz (Fig. 1b). We confirmed that these OFF RGCs retain their OFF polarity at

higher luminance by measuring their contrast response function in photopic conditions (Supplementary Fig. 1).

We stimulated OFF RGCs with drifting gratings (spatial frequency = 0.1 cycles per degree, temporal frequency = 2 Hz), at 12 different orientations to test for OS responses. Cells classified based on their light step responses by the criteria above (Fig. 1b) were strongly OS, with significantly larger OS indices (OSIs) than other OFF RGCs (Fig. 1f, two-sample $t$ test, $p < 10^{-43}$). We will henceforth refer to these RGCs as OFF OS RGCs. 95% of OFF OS RGCs (87/92) had an OSI >0.2; 4% (4/89) of other OFF RGCs exceeded this OSI threshold.

In one experiment, we encountered two OFF OS RGCs with neighboring somata (Fig. 1c). Both exhibited OS responses (Fig. 1d, e), however, their preferred angles were orthogonal ($\Delta\theta$ = 85°). Since RGCs of the same type form mosaics across the entire retina[38,39], the presence of proximate OFF OS RGCs suggested that they might be further separated into multiple, functionally distinct subtypes. We sampled the preferred angles of OFF OS RGCs and found a bimodal distribution (Hartigan's dip test $p = 0.008$, $n = 92$ cells, Fig. 1g). We classified cells with preferred orientations between [−45°, 45°] and [45°, −45°] as OFF horizontal (h) OS (nasal-temporal) and OFF vertical (v) OS RGCs (dorsoventral), respectively. We found no significant differences in light step responses, strength of orientation tuning or spatial distribution across the retina between these two OFF OS RGC types (Supplementary Fig. 2). Unless otherwise noted, we report results averaged across both OFF OS RGC types with individual data points colored green or orange for OFF vOS and OFF hOS cells, respectively.

We tested the robustness of OS in OFF OS RGCs across a range of temporal and spatial frequencies and light levels. Both vertical and horizontal OFF OS RGCs displayed strong OS responses but had somewhat different spatiotemporal tuning properties (Supplementary Fig. 3a–d). OS responses were conserved across 7 log units of light intensity (Supplementary Fig. 3e). Direction selectivity was much weaker than OS across the entire range of spatial and temporal frequencies we tested (Supplementary Fig. 3f, g). Orientation tuning was similar when OFF OS RGCs were probed with either flashed bar stimuli or drifting gratings (Supplementary Fig. 4).

We noted morphological similarity between some of the OFF OS RGCs and a previously reported RGC type, JAM-B[40] (Fig. 1c and Fig. 2). We therefore recorded from a transgenic line in which JAM-B RGCs are fluorescently labelled. All labelled RGCs recorded in this mouse line ($n = 14$ cells) were classified as OFF vOS RGCs when probed with drifting gratings (Fig. 1g), indicating that JAM-B RGCs are orientation selective. There was no significant difference between OSIs of JAM-B RGCs ($n = 14$, average OSI = 0.47 ± 0.05) and OFF vOS RGCs recorded in WT retina ($n = 38$, average OSI = 0.48 ± 0.03, $p = 0.83$, two-sample $t$ test). We return to interpret how OS might interact with the other reported functional properties of JAM-B RGCs in the Discussion.

**Morphology of OFF OS RGCs.** Perhaps the simplest mechanism by which a RGC can compute OS is by aligning its dendrites along the preferred orientation so that it receives more excitation from bipolar cells in the preferred than in the orthogonal orientation. Dendritic asymmetry is part of the mechanism of some retinal OS cells[36,37,41], but it is absent in others[34,36]. We examined the morphology of OFF OS RGCs and its relationship to their orientation preference.

Our measurements of the morphology of OFF vOS RGCs were consistent with previous reports of JAM-B RGCs[40,42]. OFF vOS RGCs were characterized by their highly asymmetric

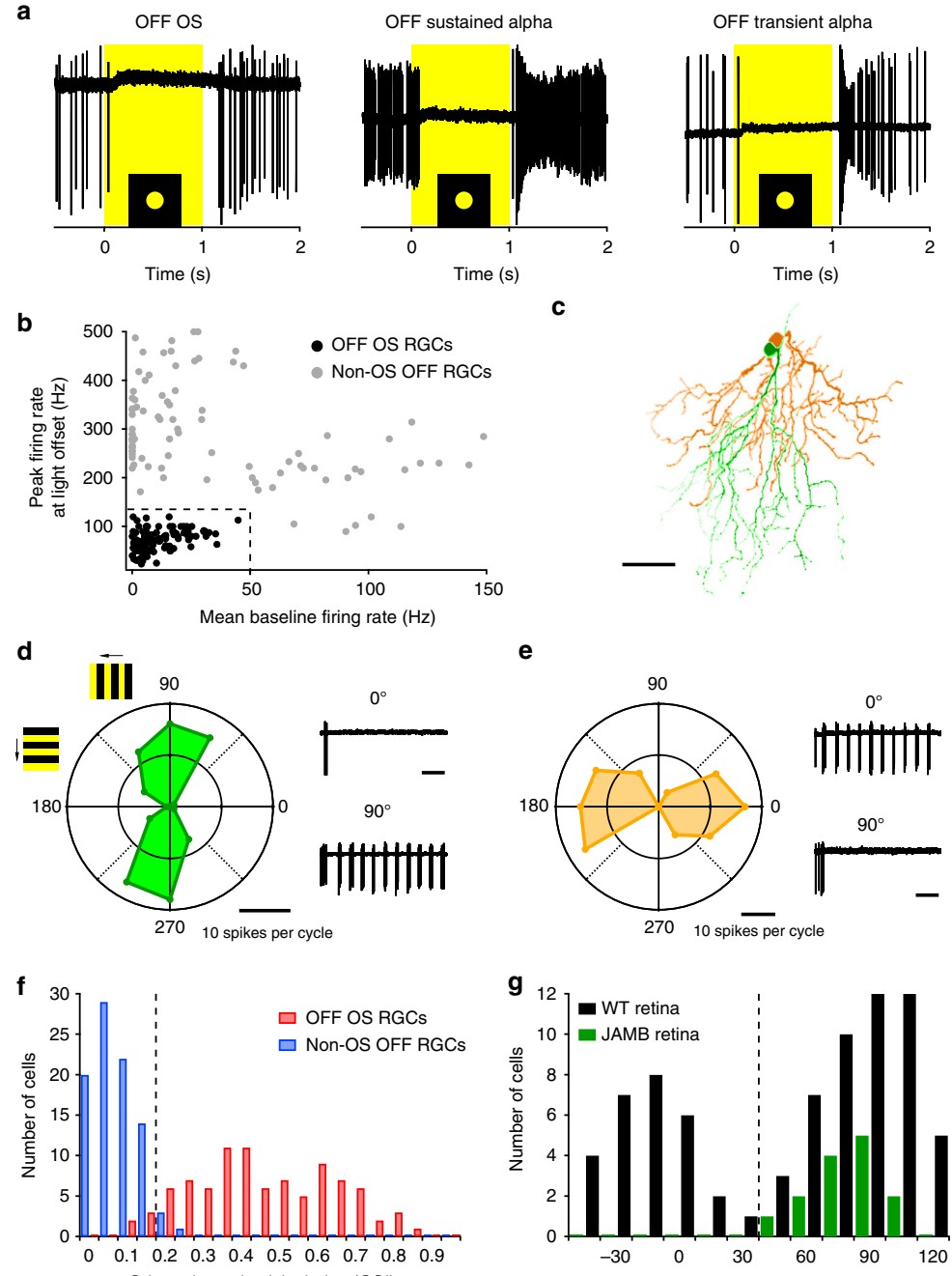

**Fig. 1** OFF OS RGC is a distinct functional cell type. **a** Responses of OFF OS (left) OFF sustained alpha (middle) and OFF transient alpha (right) RGCs to a 1 s flash of 200 μm diameter circular light spot from darkness. Yellow rectangle indicates light stimulus. **b** Peak firing rate at light offset plotted against mean baseline firing rate for OFF OS (n = 92) and non-OS (n = 84) RGCs. Dotted lines indicate peak firing rate = 135 Hz and mean baseline firing rate = 50 Hz. **c** Dendritic morphology of two neighbouring OFF OS RGCs. Scale bar = 50 μm. **d**, **e** Polar plots of drifting grating responses of OFF OS RGCs coloured in green (**d**) and (**e**) orange, respectively. F1 response amplitudes and grating angles are plotted in radial axes and angle axes, respectively. F1 cycle average responses are calculated from three trails in each angle. **f** Histogram of OSI of OFF OS (n = 92) and non-OS RGCs (n = 84) as identified by their light step response profile. Dotted line indicates OSI = 0.2. **g** Angle histogram of preferred orientations of OFF OS RGCs recorded in WT retina (black, n = 78) or labelled cells in the JAM-B line (green, n = 14). Dotted line indicates 45° angle

dendrites pointing toward ventral retina (Fig. 2a). OFF hOS RGCs displayed more symmetric dendritic morphologies than OFF vOS RGCs (Fig. 2b). However, we did encounter 2 of 22 OFF vOS RGC with symmetric dendrites. OFF vOS cells stratified distal to the OFF choline acetyltransferase (ChAT) band (Fig. 2c). OFF hOS RGCs stratified in similar locations (Fig. 2d; p = 0.56, Mann–Whitney U-test). Comparing our morphological measurements with an online database of mouse RGC morphologies

(http://www.museum.eyewire.org)[43], we believe that the most likely matches are type '2aw' for OFF vOS RGCs and type '2i' for OFF hOS RGCs.

To quantify dendritic asymmetry, we fit polygons to the dendritic trees and calculated the area and the centre of mass (COM) of the arborizations. OFF vOS RGCs had larger dendritic area than OFF hOS RGCs ($4.48 \pm 0.48 \times 10^4$ μm², n = 26 vs. $3.93 \pm 0.27 \times 10^4$ μm², n = 23, two-sample t test, p = 0.044).

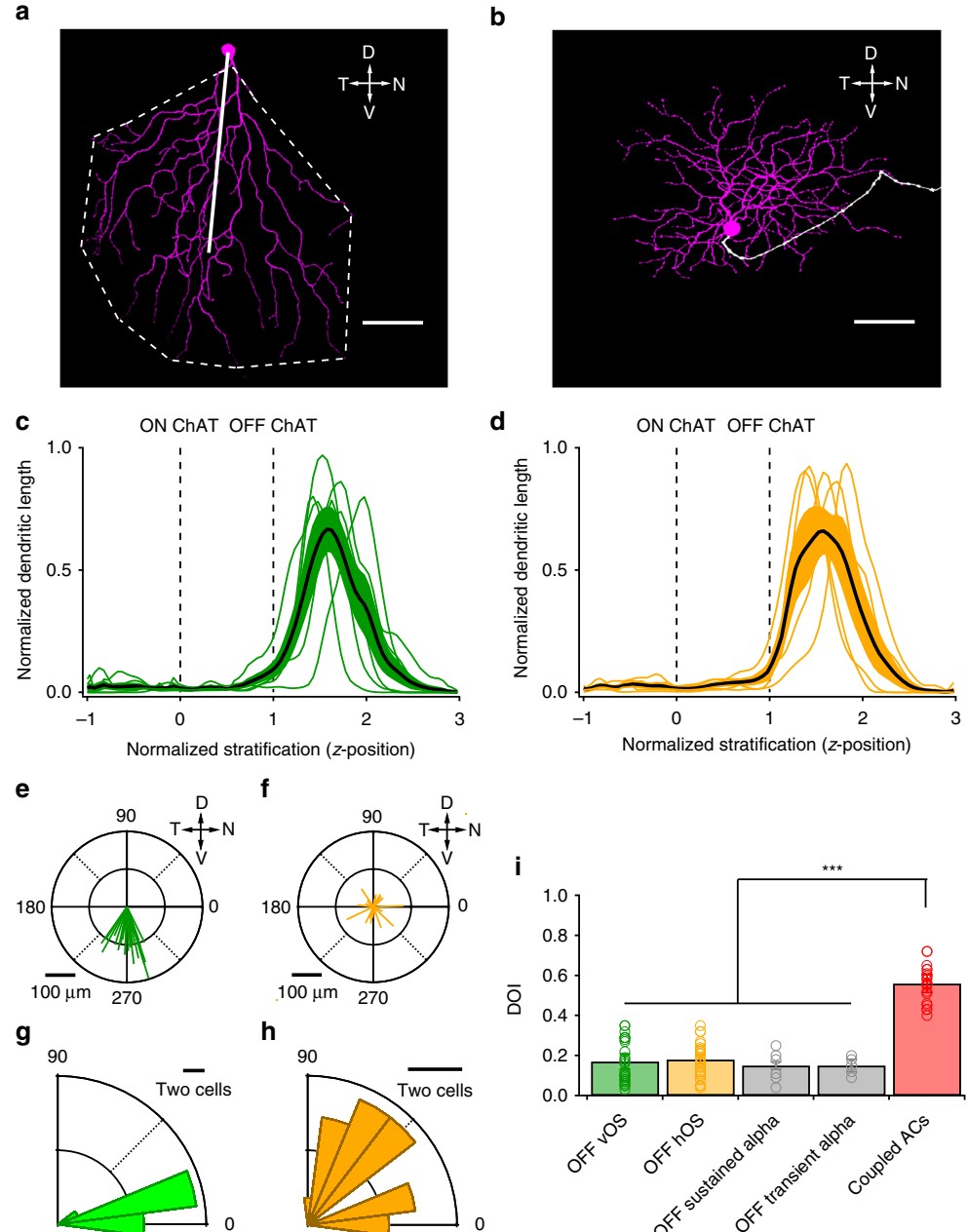

**Fig. 2** Morphology of OFF OS RGCs. **a**, **b** Dendritic morphologies of vertical OFF OS RGC (**a**) and horizontal OFF OS RGC (**b**). OFF dendrites are represented in magenta and axons are in white. Scale bar = 50 μm. Dashed line in a shows outline of dendrites. Solid line is the vector from the soma to the center of mass (COM). **c**, **d** $Z$-profiles of OFF vOS (**c**) and OFF hOS (**d**) dendritic arbour stratification. Thin green and orange lines indicate profiles of individual OFF vOS and OFF hOS cells, respectively. Thick black lines represent mean for OFF vOS ($n = 6$) and OFF hOS ($n = 5$). Dotted lines indicate ON and OFF starburst planes. The inner nuclear layer (INL) is located to the right and the ganglion cell layer (GCL) is located to the left. Shaded regions indicate SEM across cells. **e**, **f** Polar plots of COM vectors of OFF vOS ($n = 24$) (**e**) and OFF hOS ($n = 22$) (**f**) RGCs. Scale bar = 50 μm. **g**, **h** Rose plots of absolute differences between OS angle ($\theta_{OS}$) and COM vector angle ($\theta_{COM\ vector}$) of ON vOS RGCs ($n = 24$) (**g**), and OFF hOS RGCs ($n = 22$) (**h**). **i** Dendritic orientation index (DOI) of OFF OS RGCs ($n = 26$ OFF vOS and 23 OFF hOS), OFF sustained alpha ($n = 6$), OFF transient alpha ($n = 5$) and amacrine cells coupled to OFF OS RGCs ($n = 19$). ***$p < 0.001$

Vectors from the soma to the COM of the arborizations were significantly longer in OFF vOS cells than in OFF hOS RGCs ($145.52 \pm 8.09$ μm vs. $50.09 \pm 5.11$ μm, two-sample $t$ test, $p < 10^{-11}$) (Fig. 2e, f). These vectors were directed ventrally in OFF vOS RGCs ($n = 26$, $p < 10^{-6}$, Hodges–Ajne test) but pointed in random directions in OFF hOS RGCs ($n = 23$, $p = 0.13$, Hodges–Ajne test, Fig. 2e, f).

We measured the difference angle between the preferred orientation of the light response and the soma to dendritic COM vectors. In OFF vOS RGCs, the dendrites were well aligned to the preferred orientation (Fig. 2g). This relation was significantly different from a random uniform distribution (Hodges–Ajne test, $p < 0.01$). Due to the symmetric morphology of OFF hOS RGCs, there was no significant relationship between orientation preference and dendrites (Fig. 2h, Hodges–Ajne test, $p = 0.61$).

An asymmetry from soma to dendrites does not alone constitute a mechanism for OS. Since bipolar cell axons are small relative to the dendritic arbours of these RGCs and since

they form tight mosaics[44], the simple excitatory OS mechanism requires substantially oriented RGC dendrites with respect to their COM, not the soma (Discussion). We quantified the orientation of RGC dendrites relative to their COM using an index, dendritic orientation index (DOI), analogous to the vector sum orientation index we used to quantify the OS spike responses (Fig. 2i, Methods). Despite their soma-to-dendrites asymmetry, OFF vOS RGC dendrites were not significantly more oriented around their COM than those of OFF hOS RGCs or the two OFF alpha RGC types (two sample $t$ test, $p = 0.64$). Moreover, DOIs of OFF OS RGCs were similar to that of well-known OFF RGCs with symmetric dendrites (Fig. 2i, two sample $t$ test, $p > 0.2$ for all pairwise tests). In contrast with the lack of oriented dendrites in these RGCs, the coupled amacrine cells described below (see Fig. 3 and Supplementary Fig. 8) indeed had significantly higher DOI (Fig. 2i, two sample $t$ test, $p < 10^{-3}$ for all pairwise tests). These results suggest that oriented dendrites alone cannot explain OS in either of the OFF OS RGC types, but oriented morphology is likely important in certain OS amacrine cells as has been shown in the rabbit retina[37,41].

**OFF OS RGCs are coupled to ACs via Cx36 gap junctions**. In the course of our morphology experiments, we noticed coupled cell bodies after filling OFF OS RGCs with neurobiotin. We traced the dendrites of some of the cells coupled to OFF vOS RGCs. These amacrine cells had somata in the inner nuclear layer (INL) and exhibited striking asymmetric morphology (Fig. 3a). All of their dendrites were aligned dorsally, opposite to the RGC dendrites (Fig. 3a, b). The coupled ACs stratified distal to the OFF ChAT band, overlapping OFF OS RGCs (Fig. 3c). On average a single OFF vOS RGC was coupled to $19 \pm 2.3$ cells in the INL ($n = 7$ RGCs). We observed weak coupling from OFF hOS RGCs to cells in the ganglion cell layer and INL, but we were unable to characterize their morphology (Supplementary Fig. 5, Discussion).

To identify the molecular identity of the connexin proteins forming the gap junctions, we stained a coupled network with different antibodies against connexin 36 (Cx36) and connexin 45 (Cx45) (Fig. 3d–g). We found that 57% of putative connections between OFF vOS RGCs and coupled amacrine cells colocalized with Cx36 puncta. Rotating the Cx36 channel by 90° eliminated most of the colocalization. The same analysis for Cx45 revealed no significant colocalization with putative crossings (Fig. 3g). We conclude that Cx36 is the predominant (possibly the only) connexin mediating gap junctions between OFF vOS RGCs and ACs.

**Synaptic conductances in OFF OS RGCs**. To understand the functional role of gap junctions in OFF OS RGCs, we searched for an electrophysiological signature of electrical synapses. Unlike synaptic currents from chemical synapses, a source of current arising from electrical coupling remains constant in magnitude at different holding potentials and does not reverse polarity[45,46]. A 200 μm diameter spot evoked currents with both transient and sustained components at light onset. The current reversed at $-67 \pm 4$ mV (Fig. 4a, b, $n = 15$ cells), consistent with a source (mostly) from inhibitory chloride channels. Stimulation with much larger spots (1200 μm diameter) produced slower currents with only a sustained component (Fig. 4c). We took advantage of the different time course of these currents to analyse their current-voltage (IV) relationships separately in different time windows (Fig. 4a–c). Unlike the inhibitory currents for smaller spots, the magnitude of these sustained currents remained constant and did not reverse at any holding potential tested (Fig. 4d). We attributed this current to electrical synapses, and we hypothesized that it was present for small stimuli as well, but was

overwhelmed by the much larger inhibitory current. The peak outward current at light onset measured at the excitatory reversal potential was $757 \pm 76$ pA (Fig. 4f, $n = 9$ OFF vOS and OFF hOS cells). There was no significant difference between currents for OFF vOS and hOS RGCs ($p = 0.66$, Mann–Whitney $U$-test). The amplitude of gap junction currents was also similar between the two OFF OS RGC types (Fig. 4f, peak current $= 80 \pm 12$ pA, $n = 6$ OFF vOS and $72 \pm 9$ pA, $n = 5$ OFF hOS cells, $p = 0.54$, Mann–Whitney $U$-test).

To examine the spatial receptive field of the inhibition and the putative gap junction input, we presented spots of varying sizes while measuring the currents at either the excitatory or inhibitory reversal potential. These experiments revealed that inhibition acted upon narrow spatial scales, whereas the gap junction currents were more wide-field (Fig. 4e). For large spots, inhibition was absent in OFF OS RGCs, which helped us isolate the gap junction currents. Inhibition localized narrowly around the RGC dendrites is opposite to the dogma of its contribution to the receptive field surround, and we will return to this feature of our findings in the Discussion.

Pharmacology experiments revealed contributions of both GABA and glycine receptors to the inhibitory currents in OFF OS RGCs (Fig. 5a, b). GABAergic synapses account for most of the inhibition in these cells (Fig. 5c, 62% GABA, 25% glycine). In total inhibitory block, gap junction currents were still present and still showed a flat IV relationship (Fig. 5d). This result supports our interpretation of the flat IV as a sign of an electrical synapse.

Notably, we observed no excitatory current at light offset in OFF OS RGCs under our stimulus conditions. Peak currents at light offset measured at the inhibitory reversal potential were not significantly different from baseline noise before the light stimulus (Fig. 4f, $-22.08 \pm 4.74$ pA, $n = 18$ cells, $p = 0.09$, Mann–Whitney $U$-test). A lack of excitation at light offset is consistent with the spiking patterns we observed in which firing was suppressed at light onset and returned only modestly above its baseline rate at light offset (Fig. 1a).

Two recent studies reported excitatory synapses onto JAM-B RGCs in voltage-clamp[42] and by anatomical apposition with OFF bipolar cell ribbons in light microscopy[47], so we performed additional experiments in an effort to reconcile our finding of a lack of OFF excitation with these reports. First, we recorded OFF vOS RGCs in ventral retina where they have been reported to display rod–cone colour opponency involving cone-selective excitation (Supplementary Fig. 6). We recorded synaptic currents across three orders of magnitude of light intensity stimulating predominantly either short (S) or medium (M) wavelength sensitive cones. At high light levels stimulating S cones, we did observe an inward current, consistent with cone selectivity, but the IV relationship of this current remained flat (Supplementary Fig. 6a, b). Such a current with no voltage dependence is consistent with electrical synapses from an OFF amacrine cell (Figs. 3 and 4) rather than chemical excitatory synapses, which should drive a current reversing near 0 mV.

Presynaptic inhibition from amacrine cells onto bipolar cell terminals often has a powerful influence in suppressing excitatory transmission[48,49], so we analysed the currents in OFF vOS RGCs with gabazine blocking GABA$_A$ receptors throughout the retina (Supplementary Fig. 7). This pharmacological manipulation indeed revealed a small inward current at light offset. IV analysis showed a J-shaped curve reversing near 0 mV, consistent with a substantial contribution from NMDA receptors. Thus, excitatory synapses from OFF bipolar cells are present on OFF vOS RGCs, but they remained silent under the range of stimulus conditions we tested.

In summary, our voltage-clamp experiments coupled with pharmacology using inhibitory antagonists revealed three

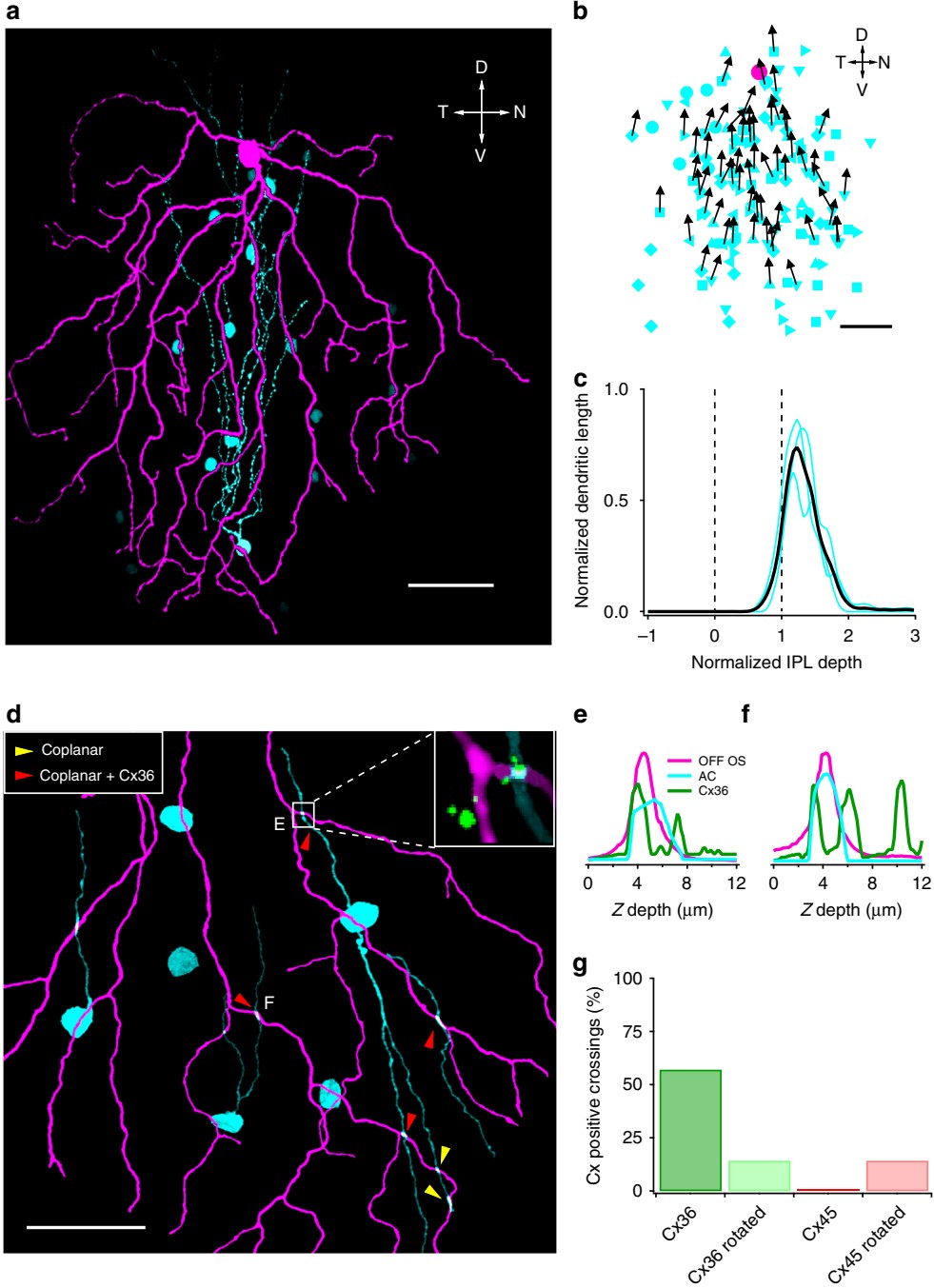

**Fig. 3** Electrical coupling between OFF OS RGCs and amacrine cells. **a** OFF OS RGC (magenta) and neurobiotin coupled amacrine cells (cyan). One of the coupled cells is traced. Scale bar = 50 μm. **b** Positions of amacrine cells coupled to OFF OS RGCs (cyan symbols). Different cyan symbols indicate cells from different experiments (n = 131 cells total from seven RGCs). Amacrine cells are plotted with respect to the RGC (magenta circle). Directions of primary dendrites are shown in black arrows when they were visible. Scale bar = 50 μm. **c** Stratification of coupled amacrine cells dendrites in IPL. Cyan lines indicate profiles of individual amacrine cells. Black line represents mean across three cells. Dotted lines indicate ON and OFF starburst planes. The inner nuclear layer (INL) is located to the right and the ganglion cell layer (GCL) is located to the left. **d** Magnified view of an OFF OS (magenta) and amacrine cell (cyan) coupled network. Yellow and red arrowheads indicate coplanar dendrite crossings and coplanar dendrite crossings positive for connexin 36 (Cx36). Inset shows an example crossing with Cx36 puncta. Scale bar = 50 μm. **e**, **f** Z-axis profiles of fluorescent intensities for OFF OS, amacrine cell dendrites and Cx36 puncta. **g** Percentage of coplanar crossings that are connexin positive. A total of 58% (4/7) coplanar crossings were Cx36 positive and 0% (0/7) were Cx45 positive. A total of 14% (1/7) coplanar crossings were positive for either Cx36 or Cx45 when the connexin channels were rotated by 90°

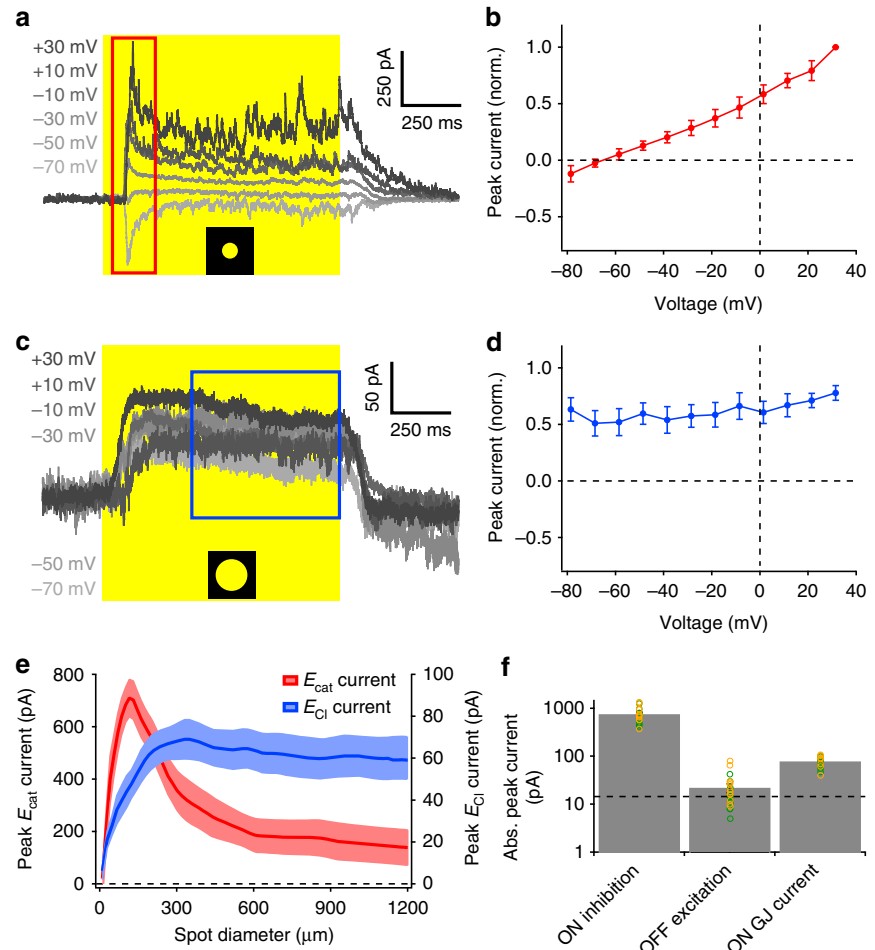

**Fig. 4** Synaptic inputs to OFF OS RGCs. **a** Current evoked in an OFF OS RGC by a light step (200 μm diameter) from darkness measured across a range of holding voltages. Yellow rectangle indicates light stimulus. **b** IV relationship for the stimulus in **a**. Error bars indicate SEM across $n = 13$ cells. Peak currents were measured in the temporal window (0–200 ms) indicated by red rectangle in **a**. **c** Current evoked in an OFF OS RGC by a light step (1200 μm diameter) from darkness measured across a range of holding voltages. Yellow rectangle indicates light stimulus. **d** IV relationship for the stimulus in **c**. Error bars indicate SEM across $n = 13$ cells. Peak currents were measured in the temporal window (500–1000 ms) indicated by blue rectangle in **c**. **e** Peak currents at excitatory reversal potential (red trace) and chloride reversal potential (blue trace) to spots of varying diameters for $n = 13$ cells. Currents were measured in a 0–200 ms temporal window at $E_{cat}$ and 500–1000 ms temporal window at $E_{Cl}$. Shaded regions indicate SEM across cells. **f** Absolute peak currents evoked by a 200 μm diameter spot of light in OFF OS RGCs. Dotted line indicates peak baseline noise level. Note logarithmic scale

currents in OFF OS RGCs: (1) inhibition at light onset driven by $GABA_A$ and glycine receptors that is strongest for small stimuli, (2) an outward gap junction current at light onset that lacks surround suppression, and (3) an excitatory current that was unmasked after $GABA_A$ receptor blockade. We interpret these results in the form of a schematic circuit diagram in the Discussion.

**Electrical synapses carry OS information**. What are the relative contributions of electrical and chemical inhibitory synapses to OS in OFF OS RGCs? We measured the tuning curves of both currents in response to drifting gratings at different orientations (Fig. 6). While inhibition was similar at all orientations, gap junction currents had larger amplitude oscillations for drifting gratings in the preferred orientation than the null orientation. Importantly, the gap junction currents in the preferred orientation included substantial fluctuations below baseline, representing a net inward current, while currents were predominantly outward in the null orientation (Fig. 6b, c). We quantified grating responses by taking the cycle average current and measuring the

peak inward current for gap junction currents and, correspondingly, the minimum outward current for inhibition. We chose this quantification because we sought to measure the influence of currents during the dark phase of the gratings, the period in which net depolarization causes OFF OS RGC's to spike in the preferred orientation. Population averages revealed OS gap junction currents and non-OS inhibition for both horizontal and vertical OFF OS RGC types (Fig. 6d–f, $OSI_{gap\ junction}$ significantly larger than $OSI_{inh}$, paired $t$ test, $p < 10^{-4}$ for both OFF vOS and hOS). The tuning of gap junction input matched that of the spiking responses (Fig. 6g).

To measure directly the coupled OS amacrine cell type implicated in this circuit, we used a transgenic mouse line (Etv1), in which several amacrine cell types are sparsely labelled (Supplementary Fig. 8). Coupling between OFF vOS RGCs and a labelled asymmetric amacrine cell was verified anatomically. Neurobiotin tracer injected in an OFF vOS RGC passed into the labelled amacrine cell, and Cx36 puncta were present at crossing points (Supplementary Fig. 8a–c). An asymmetric AC recorded in this line hyperpolarized to a light spot (Supplementary Fig. 8e), consistent with the outward gap junction currents recorded from

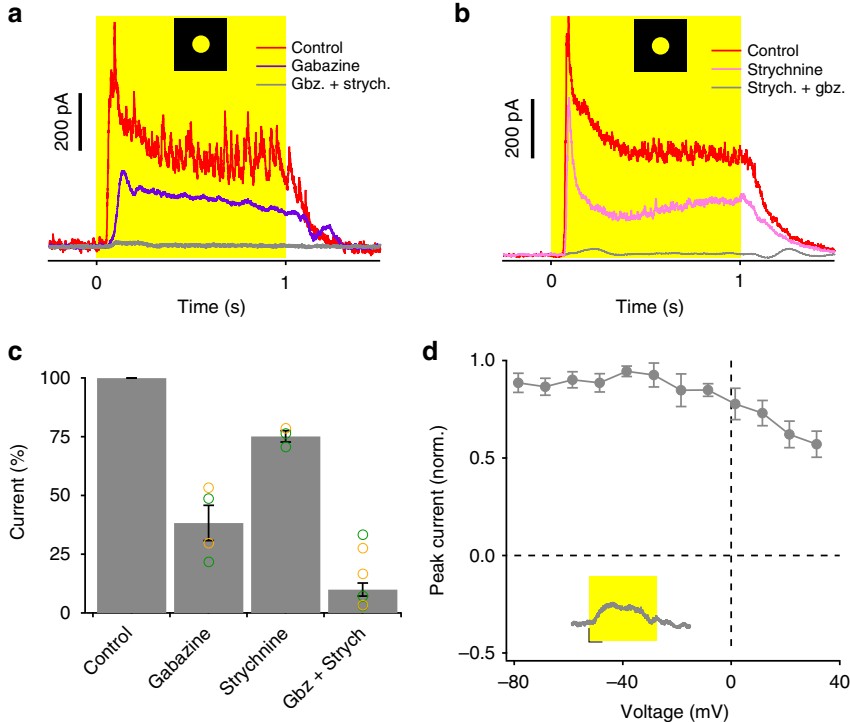

**Fig. 5** Pharmacology of inhibitory conductances in OFF OS RGCs. **a**, **b** Effect of gabazine and strychnine on light step responses (200 μm diameter) of example OFF OS RGCs from darkness. Gabazine and strychnine are bath applied first in **a**, **b**, respectively. Cells are voltage clamped at $E_{cat}$. **c** Percentage change in peak currents measured at $E_{cat}$ in inhibitory antagonists **a**, **b**. Error bars indicate SEM ($n = 4$ for gabazine, $n = 4$ for strychnine and $n = 8$ for both drugs). **d** IV relationship for the residual current in both inhibitory antagonists. Inset shows light step response (200 μm diameter) from darkness after application of both inhibitory blockers. Scale for inset: Y-axis = 100 pA, X-axis = 200 ms. Error bars indicate SEM across $n = 8$ cells

OFF vOS RGCs (Fig. 4c), and it had OS responses to drifting grating stimuli aligned to the orientation of its neurites (Supplementary Fig. 8f, g).

Next, we sought a causal manipulation to demonstrate the importance of electrical synapses in the computation of OS in OFF OS RGCs (Fig. 7). Meclofenamic acid (MFA) has been used to block gap junction-mediated signalling in the retina[50,51]. Bath application of MFA (100 μM) abolished spiking activity in OFF OS RGCs when presented with drifting gratings. We were able to record subthreshold responses in current clamp recordings. In MFA, subthreshold responses were still apparent for each cycle of the grating, but the responses were no longer OS (Fig. 7a–e, i, OSI Iclamp control = 0.33 ± 0.06, OSI MFA = 0.1 ± 0.01, p = 0.029, paired t test). While this result shows that gap junctions are necessary for the OS computation in OFF OS RGCs, it does not prove that the essential gap junctions are in the RGC itself, since MFA blocks electrical synapses throughout the retina[45]. To achieve better specificity, we blocked Cx36 gap junctions intracellularly with quinine (800 μM)[52]. We confirmed that quinine blocked the gap junction current in OFF OS RGCs with no significant effects on inhibition (Supplementary Fig. 9). When we recorded responses to drifting gratings with quinine in the patch pipette, we observed reduced spiking activity and significantly decreased OS (Fig. 7f–h, j, OSI Iclamp control = 0.33 ± 0.06, OSI quinine = 0.11 ± 0.03, p = 0.019, unpaired t test). Since our quinine manipulation was specific for the single patched RGC, and the drug is selective for Cx36 gap junctions over the other types found in the inner retina[12,52], it is likely that gap junctions essential for the OS computation were the same ones we identified anatomically (Fig. 3, Discussion). While quinine failed to eliminate all depolarization in OFF OS RGCs in response to drifting gratings possibly due to an incomplete block of gap junctions[52], the residual current was no longer OS,

providing further evidence electrical synapses, rather than chemical synapses, carry OS information to these RGCs.

**Modelling the OS computation in OFF OS RGCs.** On the basis of our measurements of the synaptic currents in OFF OS RGCs, we constructed a conductance-based leaky integrate and fire (LIF) model of the circuit (Methods). The cycle average currents measured in whole-cell voltage clamp in response to drifting gratings were used as inputs to the model, and the model generated spiking behaviour for each grating angle (Fig. 8a). Recapitulating our experimental data, the model produced more spiking in the preferred orientation than in the null orientation. The LIF model was able to capture the shape of the orientation tuning curve (Fig. 8b, c), and the preferred orientations of individual cells (Fig. 8d).

To examine the necessity and sufficiency of electrical synaptic currents in generating OS, we shuffled synaptic currents between different grating angles (Fig. 8e). For all 11 cells (7 OFF vOS and 4 OFF hOS), the LIF model achieved nearly the same degree of OS as the recorded data, even with inhibitory currents recorded at different orientations randomly shuffled (Fig. 8f, two-sample t test, p > 0.1 for all pairwise tests). In contrast, shuffling the gap junction currents largely eliminated OS in the model (Fig. 8f, two-sample t test, p < 10^{-4} for all pairwise tests). Our modelling results support the experimental result that OS in OFF OS RGCs is inherited from gap junctions with amacrine cells.

## Discussion

A schematic circuit diagram for OFF OS RGCs based on our results is presented in Fig. 9. OFF bipolar cells form excitatory synapses with OFF OS RGCs[47], however, under the range

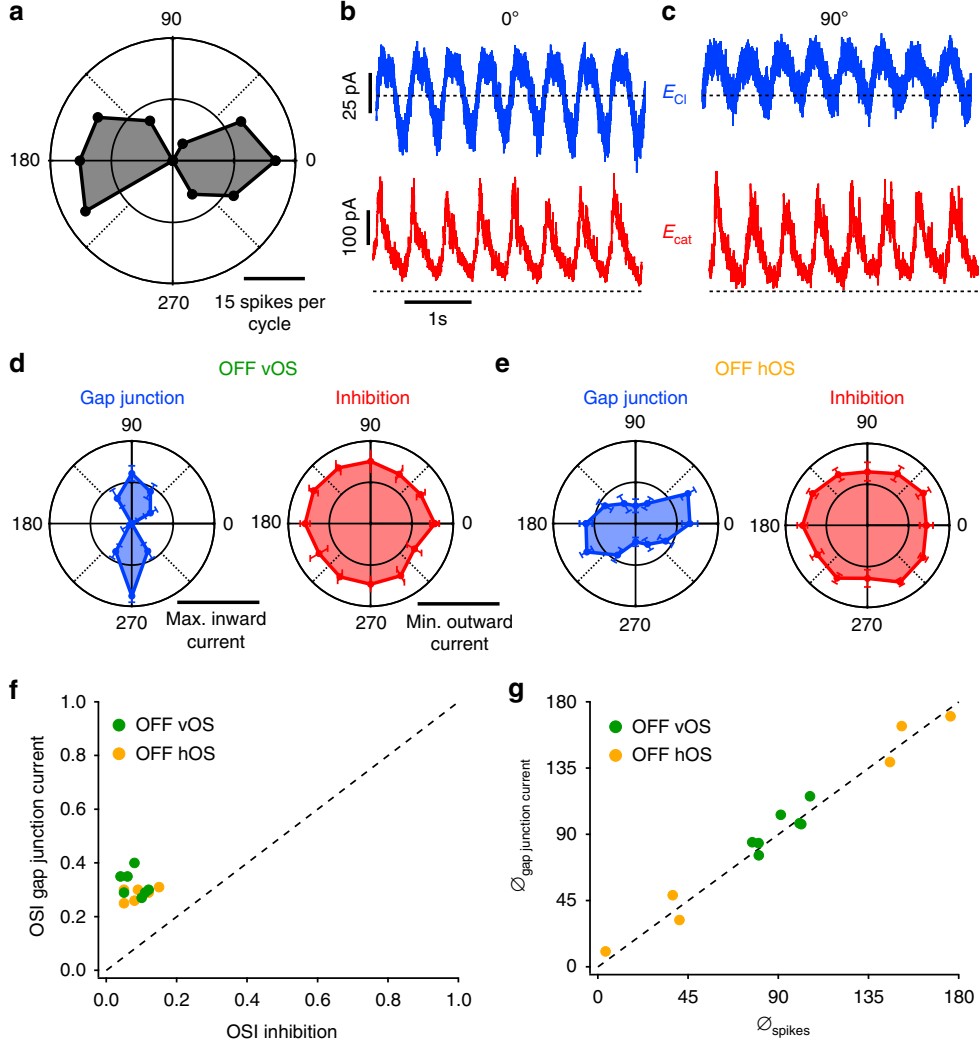

**Fig. 6** Synaptic responses to drifting gratings in OFF OS RGCs. **a** Polar plot of drifting grating responses of an example OFF OS RGC. **b**, **c** Raw traces of recorded currents at $E_{Cl}$ (blue) and $E_{cat}$ (red) currents at preferred (**b**) and null (**c**) orientations, respectively, for the cell in **a**. **d**, **e** Polar plots of population averaged normalized maximum inward gap junction (blue) and minimum outward inhibitory currents (red) for OFF vOS ($n = 7$, **d**) and OFF hOS ($n = 6$, **e**) RGCs. Error bars are SEM across cells. Inhibitory currents were obtained by subtracting the $E_{Cl}$ current from $E_{cat}$ current. **f** OSI of gap junction current plotted against OSI of inhibition for OFF vOS ($n = 7$) and OFF hOS ($n = 6$). **g** OS angle of gap junction ($\theta_{gap\ junction\ current}$) current plotted against OSI of cell attached spikes ($\theta_{spikes}$) for OFF vOS ($n = 7$) and OFF hOS ($n = 6$)

of stimulus conditions we tested, GABAergic inhibition prevents glutamate release from these synapses (Supplementary Figs. 6 and 7). At light onset, OFF OS RGCs are inhibited by both GABAergic and glycinergic narrow-field amacrine cells (Figs. 4 and 5). Additionally, wide-field (oriented) OFF amacrine cells form electrical synapses with OFF OS RGCs (Figs. 3 and 4), and these synapses carry OS information to the RGC (Figs. 6–8). The origin of OS in this circuit is presumably this coupled amacrine cell. Its highly asymmetric morphology (Figs. 2i and 3 and Supplementary Fig. 8) suggests that, like other OS amacrine cells[37,41], these cells might derive their OS simply from their asymmetric sampling of the bipolar cell mosaic (Fig. 9b), though other possibilities exist, including OS inhibition to the amacrine cell.

In this study, we report the presence of two OFF OS RGC types in the mouse retina and examine the circuit mechanisms of OS in these cells. OFF OS RGCs have been previously reported in the rabbit retina[34]. Similar to mouse, the authors reported two types of rabbit OFF OS RGCs each preferring a cardinal orientation. Vertical-preferring rabbit OFF OS RGCs possess asymmetric

dendritic morphology, whereas the horizontal-preferring RGCs have symmetric dendrites. Voltage-clamp recordings and pharmacology in rabbit OFF OS RGCs showed that OS relies on GABAergic inhibition. However, in our recordings, the combined inhibitory drive (consisting of both GABAergic and glycinergic components (Fig. 5) was found to be non-OS (Fig. 6).

Several morphological studies in teleost fish and rabbit retina report ACs similar to the ones coupled to OFF OS RGCs, all with dorsally pointing dendrites[37,52–55]. One study in rabbit found these ACs tracer coupled to a RGC type called G3[55]. On the basis of morphological similarity, they proposed that the G3 RGC in rabbit was analogous to the JAM-B RGC in mouse. The authors went on to speculate that the anatomical circuit composed of prominent asymmetric ganglion and amacrine cell dendrites might result in orientation bias. Our imaging results also revealed oriented amacrine cells coupled to OFF vOS cells (Fig. 3a–c). Furthermore, similar to our findings in mouse, gap junctions between rabbit G3 RGCs and dorsally directed ACs also contained Cx36. A comprehensive electron microscopy study[56] reported a type of amacrine cell (ac 19–30) with oriented

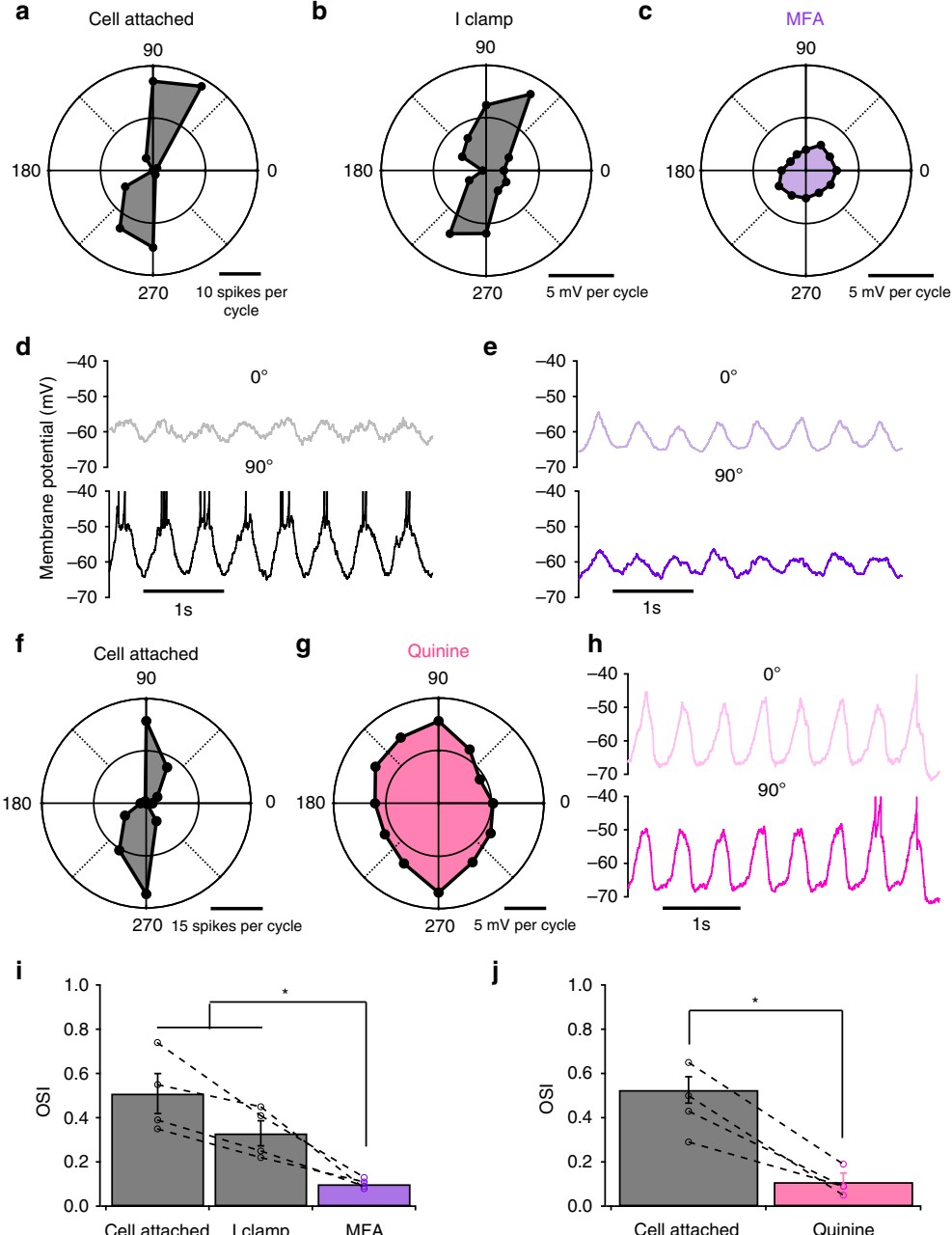

**Fig. 7** Electrical synaptic inputs are necessary for OS. **a** Polar plot of drifting grating responses of an example OFF OS RGC. **b**, **c** Polar plot of drifting grating responses of the same cell in **a** in current clamp before (**b**) and after (**c**) bath application of MFA. **d**, **e** Raw traces of recorded membrane potentials of the same cell in **a** along preferred and null orientations before (**d**) and after (**e**) bath application of MFA. **f** Polar plot of drifting grating responses of an example OFF OS RGC. **g** Polar plot of drifting grating responses of the same cell in **f** in current clamp after intracellular application of quinine. **h** Raw traces of recorded membrane potentials of the same cell in **f** along preferred and null orientations after intracellular application of quinine. **i** OSI of drifting gratings responses in cell attached mode (grey) and in current clamp mode before (grey) and after (purple) MFA application. Error bars indicate SEM across $n = 4$ ($n = 2$ OFF hOS and $n = 2$ OFF vOS) cells. Dashed lines connect measurements from the same cell. **j** OSI of drifting gratings responses in cell attached mode (grey) and in current clamp mode after quinine application (pink). Error bars indicate SEM across $n = 4$ ($n = 2$ OFF hOS and $n = 2$ OFF vOS) cells. *$p < 0.05$

dendrites and a stratification profile matching the amacrine cells we observed coupled to OFF vOS RGCs (Fig. 3a–c).

Our findings challenge the canonical organization of retinal circuits. Under a range of stimulus conditions, OFF OS RGCs lack excitatory currents. While spiking in RGCs is generally highly dependent on excitation from bipolar cells, there is precedent in the literature for RGCs in which inhibition is much larger than excitation and serves as the main driver modulating spiking[27,57,58]. We were able to unmask excitatory synapses in

GABA$_A$ receptor block (Supplementary Fig. 7), but not in any of our stimulus conditions. It remains possible that different stimuli could unmask this excitatory current, and the search for those stimulus conditions remains an interesting target for future investigation.

Another way in which the OFF OS RGC circuit differs from canonical circuits is the spatial structure of inhibition. Unlike the wide-field OS input from electrical synapses, the inhibitory currents we recorded at light onset had strong

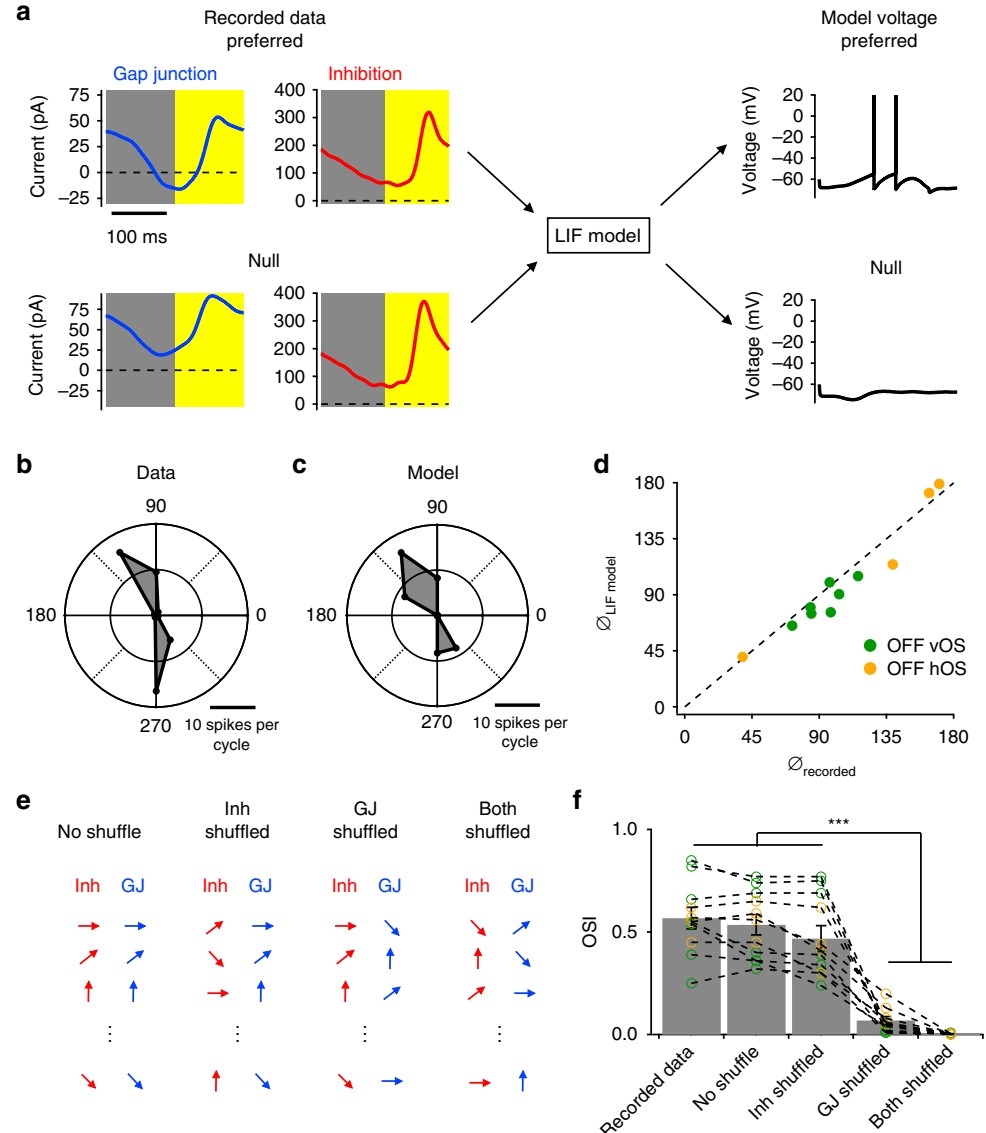

**Fig. 8** Leaky integrate and fire model of OFF OS RGCs. **a** Flowchart of the LIF model. Recorded cycle average gap junction and inhibitory currents measured in response to drifting grating stimuli were used as inputs in the LIF model to generate resulting membrane potential for different grating movement directions. **b** Polar plot of cell attached drifting grating responses of an example OFF OS RGC. **c** Simulated drifting grating response generated by the LIF model using the currents measured from the cell in **b**. **d** Scatter plot of recorded OS angles ($\theta_{recorded}$) vs. LIF model OS angles ($\theta_{LIF\ model}$). **e** Schematic showing the logic of different shuffling conditions of input currents. **f** OSI of the recorded data from four horizontal OFF OS RGCs (orange points) and seven vertical OFF OS RGCs (green points) along with OSI values from four different LIF model shuffling conditions. Dashed lines connect data and models from the same cells. Error bars indicate SEM across $n = 11$ cells. ***$p < 0.001$

surround suppression (Fig. 4e). This counters the common pattern of inhibition extending beyond the dendritic field of the RGC to contribute to its receptive field surround. Another mouse RGC was recently reported to have a similar receptive field structure in which inhibition has stronger surround suppression than excitation[59].

The most striking way in which OFF OS RGCs differ from previously described RGCs is that gap junctions, rather than playing a supportive or modulatory role[22,27], are critical for the cell's feature selectivity. Currents transmitted by electrical synapses are crucial for computing OS in OFF OS RGCs (Figs. 6–8). This represents a new paradigm for thinking about the role of amacrine cell-RGC gap junctions.

Though we observed no significant differences in the physiology or OS of gap junction currents between OFF vOS and OFF hOS RGCs, we were only able to trace the morphology

of the amacrine cells coupled to OFF vOS RGCs. Perhaps a connexin other than Cx36 is present between amacrine cells and OFF hOS RGCs, and it is less permeable to neurobiotin. The electron microscopy study described only one highly oriented amacrine cell type[56]. However, Famiglietti[55] found four basic variants of dorsally directed amacrine cells in rabbit retina. Golgi staining of one of the types showed additional neurites branches with significant deviation from vertical orientations. It is possible that such branches are selectively coupled to OFF hOS cells and provide OS inputs. Identification of the electrical synaptic partner of OFF hOS RGCs remains a target for future experiments.

There have been conflicting reports in the literature as to whether OS RGCs have dendrites aligned along the preferred orientation of the light response and whether this feature is sufficient to account for their functional OS. For a RGC with

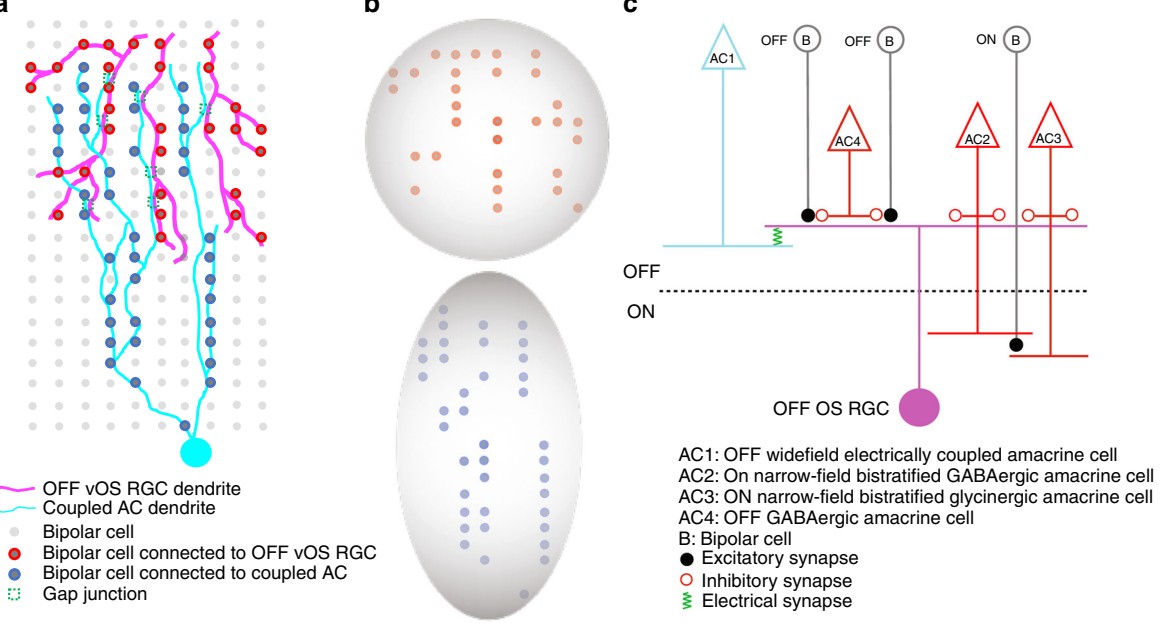

**Fig. 9** Schematic OFF OS RGC circuit model. **a** Top projection view of the OFF vOS RGC circuit. Dendrites of an OFF OS RGCs (magenta) are electrically coupled to an asymmetric amacrine cell (cyan). **b** Excitatory receptive fields for OFF vOS RGCs and coupled amacrine cells. The excitatory bipolar cell input to OFF vOS RGCs is predicted to be symmetric based on the cell's dendrites, whereas the excitatory input to the coupled cells is predicted to be asymmetric (Fig. 2). **c** Schematic model of neural circuit underlying the OS computation in OFF OS RGCs

asymmetric dendrites, the simplest mechanism by which it could achieve OS is by receiving a uniform distribution of excitatory synaptic inputs from bipolar cells along their dendritic arbours so that there is greater excitation for stimuli in the preferred orientation than for stimuli in the null orientation. In rabbit retina, both symmetric and asymmetric morphologies of OS RGCs[34,35,37,60] and OS amacrine cells[37,41] have been reported. Some studies have shown a correspondence between morphology and orientation preference[35,37,41,60], while others have not[34]. A contribution of dendritic morphology has been implicated in the computation of direction selectivity within dendritic branches of starburst amacrine cells[61,62] and in a subtype of ON-OFF DS RGCS[63]. However, another study reported no correspondence between asymmetric morphology and direction preference in DS RGCs[64]. A recent report of two additional speed-dependent DS RGCs in the mouse retina also noted the correspondence between dendritic morphology and direction preference[65].

We found that OFF vOS RGCs have asymmetric dendrites with respect to their somata, and that this asymmetry is aligned with the orientation preference of their responses (Fig. 3). This is opposite to the case of ON OS RGCs where the asymmetric morphology of ON hOS RGCs is correlated with functional preference and ON vOS RGCs possess symmetric dendrites[36]. However, several factors argue against dendritic asymmetry as a key determinant of OS in OFF OS RGCs. First, despite similar OS responses (Supplementary Fig. 2), OFF hOS RGCs displayed no dendritic asymmetry (Fig. 2). Second, even the dendrites of OFF vOS RGCs were as symmetric as those of other OFF RGCs when measured from their COM rather than from the soma (Fig. 2i). Finally, an OS mechanism based on dendritic asymmetry would presumably involve OS excitation in the RGC (Fig. 9b). We observed no excitation in OFF OS RGCs with GABAergic signalling intact (Figs. 4 and 5; Supplementary Figs. 6 and 7). Instead, OS relied on current transmitted by electrical synapses (Figs. 6–8). The functional purpose of the soma-to-dendrites asymmetry in OFF vOS RGCs remains unresolved. It may play a role in the DS that has been observed in these cells in certain

conditions[40] or it may be related to colour opponency and the dorsoventral opsin gradient[42].

Unlike OFF OS RGCs, the amacrine cells we found coupled to OFF vOS RGCs had strongly oriented dendrites (Figs. 2i and 3, Supplementary Fig. 8) in agreement with previous anatomical data from rabbit[54] and reconstructions from electron microscopy in mouse[56]. Therefore, while it appears not to be a major factor in the computation of OS in the RGCs themselves, dendritic asymmetry is likely to play a role in OS in the coupled amacrine cells as has been previously suggested in other OS amacrine cell types[37,41].

Previous reports have assigned JAM-B RGCs roles in direction selectivity, and colour opponency[40,42]. Direction selectivity was weak in our conditions (Supplementary Fig. 3f, g). Joesch and Meister[42] reported inconsistent DS in JAM-B RGCs, and DS was eliminated altogether in photopic conditions. We found that OS is consistent across cells (Fig. 1f), and stable across a large range of spatial and temporal frequencies and light levels (Supplementary Fig. 3). Though JAM-B RGCs were originally identified as DS RGCs, unlike ON-OFF and ON DS RGCs, their direction selectivity comes not from tuned synaptic input, but rather from a centre-surround receptive field asymmetry[40,42]. Thus, unlike the more robustly DS RGCs, their DS is highly dependent on properties of a moving stimulus that could affect centre-surround interactions, like colour, speed, contrast, and background luminance.

Conversely, OS in these RGCs is robust (Supplementary Fig. 3), and relies not on centre-surround interactions but on electrical synaptic input (Figs. 6–8). While it is undoubtedly an over-simplification to assign each RGC as a detector of a single feature in the visual world, we have shown that the representation of orientation is much more robust in these RGCs than that of direction. Nonetheless, JAM-B RGCs can code for orientation, direction, colour, and likely even combinations of these three features in different stimulus conditions. How multiplexed signals from RGCs are routed and decoded in the rest of the visual system is a general question of great importance for future research.

## Methods

**Animals**. Wild-type mice (C57BL/6), JAM-B CreER/Thy1-stop-YFP and Etv1-CreERT2/Ai14(tdT) transgenic mice of either sex (22 males WT, 3 males JAM-B, 1 male Etv1 and 9 females WT, 1 female JAM-B) between ages 6 weeks and 4 months were dark-adapted overnight. Animals were sacrificed following animal protocols approved by Center for Comparative Medicine (CCM), Northwestern University.

**Electrophysiology**. Retina dissections were conducted under IR light (940 nm) with assistance from IR visible light converter (night vision) goggles and separate IR dissection scope attachments (BE Meyers). Cardinal directions were identified using scleral landmarks[66]. Relieving cuts were made along cardinal directions and the whole retina was mounted ganglion cell side upon a 12-mm poly-D-lysine-coated glass coverslip (BioCoat Cellware, Corning), which was secured to a recording dish via grease. The dish was placed on the electrophysiology rig (SliceScope Pro 6000, Scientifica, UK) and superfused with carbogenated Ames medium (US Biological Life Sciences, A-1372-25; 9 mL per min) warmed to 32 °C. Tissue was illuminated at 950 nm for visualization. Cell attached recordings were obtained with a 2-channel patch-clamp amplifier (MultiClamp 700B, Molecular Devices) using pipettes (2–3 MΩ) filled with Ames solution. 2-photon illumination (960 nm, MaiTai HP, SpectraPhysics) was used for targeting fluorescently labelled somas in JAM-B transgenic mice retinas. For voltage-clamp experiments, pipettes (4–6 MΩ) were filled with an intracellular solution composed of (in mM): 104.7 Cs methanesulfonate, 10 TEA-Cl, 20 HEPES, 10 EGTA, 2 QX-314, 5 Mg-ATP and 0.5 Tris-GTP (~277 mOsm; pH ~7.32 with CsOH). To isolate excitatory and inhibitory synaptic inputs, each ganglion cell was held at the reversal potential for inhibition (~−60 mV) and excitation (~10 mV), respectively. Absolute voltage values were corrected for a liquid junction potential of −8.58 mV in the Cs-based intracellular solution. For calculation of $E_{Cl^-}$, we calculated the [Cl⁻] in Ames (105 mM) and used the Nernst equation. It was assumed that during voltage-clamp recordings, the intracellular chloride concentration was equal to the [Cl⁻] in Cs internal solution. For current clamp experiments, pipettes (4–6 MΩ) were filled with an intracellular solution composed of (in mM): 125 K-aspartate, 10 KCl, 1 MgCl₂, 10 HEPES, 1 CaCl₂, 2 EGTA, 4 Mg-ATP and 0.5 Tris-GTP (277 mOsm; pH ~7.15 with KOH). Pharmacological reagents were purchased from Sigma-Aldrich (gabazine, strychnine, MFA, quinine). Drug concentrations: strychnine, 1 μM; gabazine, 10 μM; MFA, 100 μM; quinine 800 μM.

**Visual stimuli**. A custom-designed light projection device (DLP LightCrafter, Texas Instruments) was used to display visual stimuli. All spatial stimuli patterns were displayed on a 1280 × 800 pixel array with pixel size of 2–3 μm and were focused onto the photoreceptor layer through the microscope condenser. Most experiments used blue LED illumination having peak spectral output at 450 nm. The colour experiments (Supplementary Fig. 6) used an ultraviolet LED (peak 380 nm) and green LED (peak 505 nm). Photon flux was attenuated to suitable levels using neutral density filters (Thor Labs) and light intensity values were calibrated and measured in rhodopsin isomerizations per rod per second (R*/rod/s) or cone isomerizations (S* or M*/cone/s). During cell attached recordings, the RGC's response to horizontal and vertical bars (200 × 40 μm) across 13 locations along each axis spaced by 40 μm were measured to obtain the spatial position of receptive field (RF) center. Subsequent stimuli were delivered at the RF center. Circular spots of 200 μm diameter on dark background were used to identify light step profiles of RGCs. Spots of diameters ranging from 10–1200 μm were used to characterize the spatial dynamics of RGC responses. Moving bar stimuli consisted of rectangular bars (600 × 50 μm) moving at 1000 μm/s for 3 s across the RF of RGCs. All such stimuli were presented at 200 R*/rod/s. Full-field sine wave drifting gratings are presented from a background intensity of 500 R*/rod/s at a Weber contrast of 100% for 5 s to identify OFF OS RGCs. Spatial and temporal frequencies of drifting gratings are varied between 0.025, 0.05, 0.1 and 0.2 cycles per degree (cpd) under the approximation of 1 cycle per 30 microns on the retina, and 1, 2, 4, and 8 Hz, respectively. Dark bars were flashed from a mean illuminance level of 500 R*/rod/s. All stimuli with varying parameters were presented in pseudorandom order.

**Immunohistochemistry**. Tissues were fixed for 30 min in 4% paraformaldehyde (Electron Microscopy Sciences) and incubated in 0.1 M phosphate buffer (PB) overnight at 4 °C. Fixed retinas were incubated in PBS containing 3% normal donkey serum (blocking agent), 0.05% sodium azide, 0.5% Triton X-100 for 2 h. This was followed by incubation in blocking solution and primary antibody against ChAT (Millipore, AB144P, goat anti-ChAT, 1:50 v/v) for five nights at 4 °C. Afterwards, tissues were rinsed in 0.1 M PB and incubated for two nights at 4 °C with secondary antibody against goat IgG (Jackson ImmunoResearch, 711-605-152, donkey anti-goat Alexa 647, 1:500 v/v) and streptavidin (Thermo Scientific, DyLight 488, 1:500 v/v). Following immunostaining, retinas were mounted on slides with Vectashield Antifade mounting (Vector Labs) medium.

**Imaging**. Prior to whole-cell recordings, patch pipettes were filed with AlexaFluor 488 or AlexaFluor 568. After recording, RGC morphology was imaged using two photon microscopy (920 or 760 nm, MaiTai HP, SpectraPhysics) under a ×60 water immersion objective (Olympus LUMPLan FLN 60x/1.00 NA). Emission was collected by a 520–540 nm bandpass filter.

For dendritic stratification, target RGCs were injected via patch pipettes containing Neurobiotin tracer (Vector Laboratories, SP-1150, ~3% w/v and ~280 mOsm in potassium aspartate internal solution). To improve our ability to resolve the morphology of the coupled cells, we maintained the preparation in bright light (~10⁶ R*/rod/s) for 4 h after the neurobiotin fill before fixation. Light has been reported to increase coupling among RGCs and between RGCs and amacrine cells[67].

Fixed tissues were imaged on a Nikon A1R laser scanning confocal microscope mounted on a Nikon Ti ZDrive PerfectFocus microscope stand equipped with an inverted ×40 and ×60 oil immersion objective (Nikon Plan Apo VC ×40/×60/1.4 NA). RGC dendrites and ChAT labelling were imaged at 488 and 647 nm excitation, respectively. All confocal images were collected with spacing of 0.2 μm in the z-axis. Dendritic arbours were traced using Fiji software Simple Neurite Tracer plugin. For dendritic stratification profiles similar program and methods were used as described in ref. [68].

**Data analysis**. The baseline firing rates of OFF OS RGCs and other OFF RGCs were calculated in a 500 ms time window before a light step and averaged across 10 epochs. Peak firing rates at light offset were calculated from peri-stimulus time histograms of light step responses. OSI, direction selectivity index and preferred orientation and direction angles were calculated based on a standard metric of the circular variance. Vector sum of the responses across orientations is given by:

$$\frac{\sum R(\theta)e^{2i\theta}}{\sum R(\theta)} \text{ for OSI and } \frac{\sum R(\theta)e^{i\theta}}{\sum R(\theta)} \text{ for DSI,}$$

where $R(\theta)$ is the response for 'θ' orientation across the entire stimulus time window for both drifting gratings. Absolute amplitude gives the value of this index and the phase of the resultant complex number (or half the phase) gives the value of preferred direction/orientation.

For computing COM, the RGC dendritic field was fitted with a polygon using a custom written Matlab script (http://www.github.com/SchwartzNU/SymphonyAnalysis). The polygon perimeter was resampled into 1000 points and the centroid or COM was computed. Vectors were constructed from the COM to the RGC somas and COM vector lengths and angles were measured with respect to the soma as the origin. For measuring dendritic orientation index, vectors were constructed from the centroid to the perimeter points and the vector sum calculated similarly using the above equation. Half of the complex phase of the sum gives the preferred orientation of the dendrites.

Population averaged IV curves were generated by normalizing currents across all voltages to the maximum current for each cell followed by averaging across cells. For population averages of maximum inward and minimum outward currents, the currents were normalized by the absolute value of the currents and then averaged across cells.

Spikes in current clamp recordings were removed using a sliding window average (25 ms window, 250 data points) and subthreshold oscillations were analysed subsequently.

All electrophysiological data were analysed with a custom open-source Matlab analysis package (http://www.github.com/SchwartzNU/SymphonyAnalysis), and figures were constructed in Igor 6.36 (Wavemetrics, Portland, OR).

**Modelling**. To model the membrane voltage response of OFF OS RGCs to drifting gratings, a leaky integrate and fire (LIF) neuronal model was used (Fig. 8). The subthreshold voltage across the membrane is given by the differential equation:

$$\tau \frac{dV}{dt} = (V_{rest} - V(t) - RI_{ext}(t)),$$

where V is the membrane voltage, $V_{rest}$ the resting potential, R the leak resistance and τ is the membrane time constant. $I_{ext}(t)$ represents the 'external' current, which is equal to the sum of excitatory and inhibitory synaptic currents and is given by

$$I_{ext}(t) = I_{gj} + (V(t) - V_{rev,inh})g_{inh}(t),$$

where $I_{gj}$ is the gap junction current and $V_{rev,inh}$ and $g_{inh}(t)$ are the inhibitory reversal potential and synaptic conductance respectively. The inhibitory conductance was calculated by dividing currents measured in voltage clamp by the difference between the clamped voltage and the inhibitory reversal potential. It was assumed in this model that all inhibitory currents are carried by Cl⁻ ions. Spiking is incorporated into the model by resetting the voltage to a constant value $V_{reset}$ once it has crossed a threshold $V_{th}$. Following a spike, τ was increased temporarily ($\tau_{refractory}$) to mimic a refractory period. Cycle average gap junction and inhibitory currents were randomly permuted between different grating angles and bootstrapped 10,000 times using a custom written MATLAB script to generate an average OSI for shuffling conditions.

**Statistical analysis**. Data sets were compared using a Student's *t* test (two-sided), Wilcoxon signed rank test or the Mann–Whitney *U*-test, Hodges–Ajne test, Hartigan's dip test with the test selected according to data structure. Statistical significance was accepted at $p < 0.05$. No statistical methods were used to predetermine sample sizes. Data collection and analysis were randomized or performed blind to the conditions of the experiments. Numerical values are presented as mean $\pm$ s.e.m.

**Data availability**. The data sets generated during and/or analysed during the current study are available from the corresponding author on reasonable request.

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

## Acknowledgements

We are thankful to all Schwartz lab members for helpful discussion and Susan Lynn Wohlgenant for technical support. We would like to acknowledge T.M. Schmidt, W.N. Grimes and S.P. Kuo for reviewing this manuscript and providing important feedback. Funding for this research comes from a Research to Prevent Blindness Career Development Award, National Institutes of Health Grant DP2-DEY026770A and a Karl Kirchgessner Foundation Vision Research Award.

## Author contributions

A.N. and G.W.S. designed the research; A.N. and G.W.S. performed the experiments; A.N. and G.W.S. analysed the data; and A.N. and G.W.S. wrote the paper.

## Additional information

**Competing interests:** The authors declare no competing financial interests.

