## [Peer Review File · Nature Communications]

Reviewers' comments:

Reviewer #1 (Remarks to the Author):

Orientation selectivity is one of the most prominent features in the early visual system, and first arises in the retina. However, the synaptic mechanisms underlying retinal orientation selectivity are not well understood. In this study, the authors characterized the receptive field properties and the underlying synaptic mechanisms of a type of Off orientation selective retinal ganglion cells (Off OS RGCs) in the mouse retina. They found that orientation selectivity of these Off OS cells is due to gap junction-mediated inputs from coupled amacrine cells, but not from chemical synaptic inputs. The authors used a combination of techniques including patch-clamp electrophysiology, anatomy, pharmacology and modeling. The experimental data are of high quality and convincing. Their findings fill important knowledge gaps in the retinal orientation selective circuit and electrical synapse function.

Specific comments

1. The firing of OFF OS RGCs is strongly suppressed upon light onset during a flash of a bright circular spot (Figure 1a). The Off response following the offset of the spot appears quite weak and sustained. How long does the Off response last? In addition to the quantification of the peak firing rate, it would be informative to show spike histograms covering the time before, during and after the bright spot stimulus. Responses to a dark spot should also be included.
2. While the Off OS RGCs are suppressed by the bright spot, their responses to flashed oriented bars show a pronounced increase upon light onset, particularly along the preferred orientation (Supplementary Figure 3). The Off response during this bright bar stimulus appears to be absent. As a result, the cell looks like a typical On cell. Why are the spiking patterns under these two conditions so different? Stimulus-dependent polarity has been previously reported in RGCs (e.g. Pearson & Kerschensteiner 2015, Tikidji-Hamburyan et al. 2014), however a flashing spot and a flashing bar are very similar. This unusual phenomenon should be discussed. For this reason, whole-cell voltage clamp recording during flashing oriented bars would be helpful to include in the manuscript. Also, recordings using dark bars (as in Venkataramani & Taylor, 2010) would be informative to compare with recordings from the rabbit retina.
3. When GABA signaling is intact, no excitatory synaptic currents from bipolar cells (Figure 4) are detected during a flashing spot stimuli and that quinine blocks the excitatory gap junction current (Supplementary Figure 6). However, in current clamp mode even in the presence of gap junction blockers (MFA and quinine), OFF OS RGCs are still depolarized during drifting gratings (Figure 7). What is the source of this depolarization? Does motion reveal differential synaptic contributions compared to stationary stimuli? Whole cell voltage clamp recordings during drifting gratings in the presence of quinine would reveal if in fact motion evokes an excitatory component that is otherwise masked.

In addition, the authors should correct some minor mistakes in the manuscript:

1. Some concentrations in the manuscript were reported in "μm", these should be changed to "μM".
2. In supplementary Figure 6, the caption in (c) says a flashing spot of 1200 μm was used but the diagram depicts a small spot corresponding to 200 μm.
3. Page 5, line 222, missing value for peak current for Off hOS cells.

Reviewer #2 (Remarks to the Author):

This manuscript reports the contribution of gap junctions to orientation selective light responses in retinal ganglion cells (RGCs). It studies two types of mouse RGCs, one being a previously reported type of cell, the jam-b cell, which has been reported to have direction selective and color opponent properties. The key new findings are the apparent input from electrical synapses, and their requirement for orientation selectivity. The manuscript further reports chemical inhibitory input. An additional claim is made as to the lack of chemical synaptic input, but this claim has difficulties. The finding that detection of a visual feature arises from gap junction input is novel and interesting, and would be appropriate for publication if the one main issue was resolved.

Major points.

1. The claim of the lack of chemical excitatory input is not supported well. If there are strong inhibitory conductances and gap junction conductances, it is hard to see how one can conclude that there is not chemical excitatory input given the reported IV curves. In addition, it is not clear how some contribution from nonlinear NMDA conductances can be ruled out. Finally, because they report that chemical excitatory input can be observed when inhibition is blocked, the claim that these neurons 'lack chemical excitatory input under normal physiological conditions' implies there is an unused input, which is a further strange claim. And this claim must only be restricted to the spot stimuli that it is based on.
2. Furthermore, the claim about the lack of chemical excitation is unnecessary to the main findings of the paper, and in the best case this claim can only be limited to the specific stimuli tested. The paper would stand without this claim. It is much easier to support an affirmative observation than a claim that something does not exist. This paper could be remembered for identifying the contribution of electrical synapses to orientation selectivity, or for making the mistake about the lack of chemical excitation in jam-b cells.

Reviewer #3 (Remarks to the Author):

The authors describe two OFF orientation-selective cell types in the mouse retina. This work is complementary to recently characterized ON orientation-selective neurons described by Nath and Swartz, 2016. In an analogous manner to that paper the authors describe:

1. How to putatively identify orientation-selective neurons using a flashed spot.
2. Each neurons' orientation-selective response properties

3. Each neuron's anatomy
4. Each neurons synaptic inputs.

Interestingly they find that, at least, one of the orientation-selective neurons, the vertical orientation-selective neurons, was strongly electrically coupled with a set of asymmetric OFF amacrine cells. A combination of voltage clamp experiments and pharmacology suggest that under normal conditions the main excitatory drive to these neurons arrives via these electrically coupled amacrine cells. The anatomy of these neurons is extremely elongated, suggesting a possible substrate of the ganglion cell's orientation selectivity. They are able to demonstrate that the gap junction mediated, but not the inhibitory, inputs to each orientation selective neuron are orientation selective. Blocking of electrical synapses blocked most of the excitatory drive to the recorded retinal ganglion cells, suggesting that the orientation selective responses are driven by input from the electrically coupled orientation selective amacrine cells.

Taken together this work makes a convincing case that input via electrically coupled amacrine cells underlie the orientation selective signal driving vertically orientated cells. My main concern is that the pharmacological blockage of electrical synapses also blocks most of the excitatory drive to the orientation-selective ganglion cells. Having not recorded from the putatively orientation selective amacrine cells makes it difficult to nail down the circuitry.

Major Comments.

1. The blockage of electrical synapses also blocks the major driving force to the neuron, making this result somewhat trivial in assessing their role in orientation selective response of the ganglion cell.
2. As the anatomy of the retinal ganglion cell is a poor predictor of its response property, the same assumption should be made for the putatively orientation selective amacrine cells. More evidence needs to be provided that these neurons responses are indeed orientation selective. Recording from these neurons directly would greatly enhance the impact of the paper. Alternatively, a more fine grained analysis of their input currents may be sufficient. Given the apparent small size of the amacrine cells (Fig 3), is the local input from electrical synapses orientation selective?
3. Figure 4. It needs to be better explained why the polarity of recorded putative gap junctional currents are outward. It would be easier to follow the logic of the experiments if the polarity of the stimulus matched the polarity of the response of the neuron, light decrements. Otherwise some clarification needs to be provided to make the logic of the arguments easier to follow.
4. Related to point 3. In Supplementary Figure 4 a putative excitatory current is shown. Why does its magnitude not change and reverse with the holding potential? According to the main text (line 207) this has all the characteristics of a gap junction mediated current. Is the title of this figure correct?
- 5.

Minor Comments.

1. On line 333 it states that wide-field OFF amacrine cells form electrical synapses. In Figure 3 and 4 these neurons appear smaller than the ganglion cells. Wide-field seems a misleading name. Please use language that is not misleading.
2. Need to be more specific when results only apply to vertical or horizontal orientation selective neurons. The grouping of results strongly suggests the model generalizes to both horizontal and vertical orientation selective neurons. The evidences presented, although

suggestive, does not adequately support this generalization.

3. In the analysis of input currents in Figure 4e it is unclear whether you are analyzing the early or late phase of the response or what spot sizes were presented.

4. Supplementary Figure 3. A-c and b-d do not appear to be the same cells. The illustration of bar angle in a does not match x-axis.

We thank the reviewers for their positive and thoughtful comments. We believe that answering their concerns has improved the paper. Detailed responses are below, and corresponding changes in the main text are highlighted in red.

Reviewer #1 (Remarks to the Author):

Orientation selectivity is one of the most prominent features in the early visual system, and first arises in the retina. However, the synaptic mechanisms underlying retinal orientation selectivity are not well understood. In this study, the authors characterized the receptive field properties and the underlying synaptic mechanisms of a type of Off orientation selective retinal ganglion cells (Off OS RGCs) in the mouse retina. They found that orientation selectivity of these Off OS cells is due to gap junction-mediated inputs from coupled amacrine cells, but not from chemical synaptic inputs. The authors used a combination of techniques including patch-clamp electrophysiology, anatomy, pharmacology and modeling. The experimental data are of high quality and convincing. Their findings fill important knowledge gaps in the retinal orientation selective circuit and electrical synapse function.

Specific comments

1. The firing of OFF OS RGCs is strongly suppressed upon light onset during a flash of a bright circular spot (Figure 1a). The Off response following the offset of the spot appears quite weak and sustained. How long does the Off response last? In addition to the quantification of the peak firing rate, it would be informative to show spike histograms covering the time before, during and after the bright spot stimulus. Responses to a dark spot should also be included.

Thank you for this suggestion. We have added a supplementary figure (Supplementary Fig. 1) that includes both PSTHs for a dark spot and a population average contrast response function.

2. While the Off OS RGCs are suppressed by the bright spot, their responses to flashed oriented bars show a pronounced increase upon light onset, particularly along the preferred orientation (Supplementary Figure 3). The Off response during this bright bar stimulus appears to be absent. As a result, the cell looks like a typical On cell. Why are the spiking patterns under these two conditions so different? Stimulus-dependent polarity has been previously reported in RGCs (e.g. Pearson & Kerschensteiner 2015, Tikidji-Hamburyan et al. 2014), however a flashing spot and a flashing bar are very similar. This unusual phenomenon should be discussed. For this reason, whole-cell voltage clamp recording during flashing oriented bars would be helpful to include in the manuscript. Also, recordings using dark bars (as in Venkataramani & Taylor, 2010) would be informative to compare with recordings from the rabbit retina.

We sincerely apologize for our error in Supplementary Fig. 3 that caused this confusion about response polarity. The flashed bar stimuli were indeed dark flashed bars at different orientations. We included the wrong pictogram of the stimulus in the previous

version. OFF OS RGCs respond to the onset of dark bars with increased firing as expected from an OFF cell. No stimulus dependent polarity switch is observed in these RGCs and this issue has been addressed in the updated figure. The addition of Supplementary Fig. 1, showing the contrast response function, provides further evidence that OFF OS RGCs retain OFF polarity in photopic conditions.

3. When GABA signaling is intact, no excitatory synaptic currents from bipolar cells (Figure 4) are detected during a flashing spot stimuli and that quinine blocks the excitatory gap junction current (Supplementary Figure 6). However, in current clamp mode even in the presence of gap junction blockers (MFA and quinine), OFF OS RGCs are still depolarized during drifting gratings (Figure 7). What is the source of this depolarization? Does motion reveal differential synaptic contributions compared to stationary stimuli? Whole cell voltage clamp recordings during drifting gratings in the presence of quinine would reveal if in fact motion evokes an excitatory component that is otherwise masked.

This is an astute observation. We added a sentence in the text to address this point (lines 347-350). We cannot be sure about the source of depolarization in the presence of quinine or MFA, but part neither drug provides a complete block^{1,2}, so one possible explanation is a bit of residual gap junction current. Alternatively, the reviewer may be correct that motion stimuli engage some excitatory circuit. Throughout the paper, we have removed our claim that OFF OS RGCs receive no excitation and replaced it with the more thoroughly supported claim that excitation does not carry OS information to these cells. The MFA and quinine experiments support this conclusion, because even if the residual depolarization does involve excitation, that excitatory component is not OS.

In addition, the authors should correct some minor mistakes in the manuscript:

1. Some concentrations in the manuscript were reported in “ μm ”, these should be changed to “ μM ”.
2. In supplementary Figure 6, the caption in (c) says a flashing spot of 1200 μm was used but the diagram depicts a small spot corresponding to 200 μm .
3. Page 5, line 222, missing value for peak current for Off hOS cells.

All these errors have been fixed. Thank you.

Reviewer #2 (Remarks to the Author):

This manuscript reports the contribution of gap junctions to orientation selective light responses in retinal ganglion cells (RGCs). It studies two types of mouse RGCs, one being a previously reported type of cell, the jam-b cell, which has been reported to have direction selective and color opponent properties. The key new findings are the apparent input from electrical synapses, and their requirement for orientation selectivity. The manuscript further reports chemical inhibitory input. An additional claim is made as to the lack of chemical synaptic input, but this claim has difficulties. The finding that detection of a visual feature arises from gap junction input is novel and interesting, and would be appropriate for publication if the one main issue was resolved.

Major points.

1. The claim of the lack of chemical excitatory input is not supported well. If there are strong inhibitory conductances and gap junction conductances, it is hard to see how one can conclude that there is not chemical excitatory input given the reported IV curves. In addition, it is not clear how some contribution from nonlinear NMDA conductances can be ruled out. Finally, because they report that chemical excitatory input can be observed when inhibition is blocked, the claim that these neurons 'lack chemical excitatory input under normal physiological conditions' implies there is an unused input, which is a further strange claim. And this claim must only be restricted to the spot stimuli that it is based on.
2. Furthermore, the claim about the lack of chemical excitation is unnecessary to the main findings of the paper, and in the best case this claim can only be limited to the specific stimuli tested. The paper would stand without this claim. It is much easier to support an affirmative observation than a claim that something does not exist. This paper could be remembered for identifying the contribution of electrical synapses to orientation selectivity, or for making the mistake about the lack of chemical excitation in jam-b cells.

These are excellent points, and we agree that proving the negative that OFF OS RGCs lack all excitatory input is both not well supported and not essential to our story. Throughout the manuscript (including that sentence in the abstract), we have removed our claim that OFF OS RGCs receive no excitatory input and replaced it with the well supported claim that excitation does not contribute to the OS computation in these cells.

Reviewer #3 (Remarks to the Author):

The authors describe two OFF orientation-selective cell types in the mouse retina. This work is complementary to recently characterized ON orientation-selective neurons described by Nath and Schwartz, 2016. In an analogous manner to that paper the authors describe:

1. How to putatively identify orientation-selective neurons using a flashed spot.
2. Each neurons' orientation-selective response properties
3. Each neuron's anatomy
4. Each neurons synaptic inputs.

Interestingly they find that, at least, one of the orientation-selective neurons, the vertical orientation-selective neurons, was strongly electrically coupled with a set of asymmetric OFF amacrine cells. A combination of voltage clamp experiments and pharmacology suggest that under normal conditions the main excitatory drive to these neurons arrives via these electrically coupled amacrine cells. The anatomy of these neurons is extremely elongated, suggesting a possible substrate of the ganglion cell's orientation selectivity. They are able to demonstrate that the gap junction mediated, but not the inhibitory, inputs to each orientation selective neuron are orientation selective. Blocking of electrical synapses blocked most of the excitatory drive to the recorded retinal ganglion cells, suggesting that the orientation selective responses are driven by input from the electrically coupled orientation selective amacrine cells.

Taken together this work makes a convincing case that input via electrically coupled amacrine cells underlie the orientation selective signal driving vertically orientated cells. My main concern is that the pharmacological blockage of electrical synapses also blocks most of the excitatory drive to the orientation-selective ganglion cells. Having not recorded from the putatively orientation selective amacrine cells makes it difficult to nail down the circuitry.

Major Comments.

1. The blockage of electrical synapses also blocks the major driving force to the neuron, making this result somewhat trivial in assessing their role in orientation selective response of the ganglion cell.

We respectfully disagree regarding the importance of this result. If the cell had relied on OS excitation rather than gap junctions, blocking electrical synapses (especially internally with quinine) would not have removed the major driving force of the neuron. The fact that it did so supports our claim that electrical synapses drive OS in these cells. Additionally, we performed the experiments in current clamp, rather than cell attached, so that we could measure the OS of subthreshold voltages even if our manipulations eliminated spiking. Subthreshold OS was significantly reduced in the presence of quinine relative to control conditions (Fig. 7i, j).

2. As the anatomy of the retinal ganglion cell is a poor predictor of its response property, the same assumption should be made for the putatively orientation selective amacrine cells. More evidence needs to be provided that these neurons responses are indeed

orientation selective. Recording from these neurons directly would greatly enhance the impact of the paper. Alternatively, a more fine grained analysis of their input currents may be sufficient. Given the apparent small size of the amacrine cells (Fig 3), is the local input from electrical synapses orientation selective?

We have performed current clamp recordings from one of these amacrine cells (ACs) and provided evidence that they are indeed orientation selective, aligned with their dendrites (see Supplementary Fig. 8). The apparent small size of these ACs (Fig. 3) is due to the lack of permeation of neurobiotin tracer to the distal dendrites. After patching these ACs with Alexa488 dye we observed wide-field morphology (Supplementary Fig. 8d).

3. Figure 4. It needs to be better explained why the polarity of recorded putative gap junctional currents are outward. It would be easier to follow the logic of the experiments if the polarity of the stimulus matched the polarity of the response of the neuron, light decrements. Otherwise some clarification needs to be provided to make the logic of the arguments easier to follow.

The amacrine cell recordings help clarify this. In response to a circular light spot from darkness the AC hyperpolarizes (Supplementary Fig. 8e). Hence, the gap junction current recorded in the OFF OS RGC at light onset is outward. Interestingly, the amacrine cell does depolarize above baseline for drifting gratings in the preferred orientation (Supplementary Fig. 8g) consistent with the inward current we observed for the same stimulus in the RGC at the inhibitory reversal potential (Fig. 6b).

4. Related to point 3. In Supplementary Figure 4 a putative excitatory current is shown. Why does its magnitude not change and reverse with the holding potential? According to the main text (line 207) this has all the characteristics of a gap junction mediated current. Is the title of this figure correct?

We apologize for the incorrect figure title. This has been fixed now in Supplementary Fig. 6.

5.
Minor Comments.

1. On line 333 it states that wide-field OFF amacrine cells form electrical synapses. In Figure 3 and 4 these neurons appear smaller than the ganglion cells. Wide-field seems a misleading name. Please use language that is not misleading.

Please see Supplementary Fig. 8d. The OFF amacrine cells have a dendritic arbor length of ~500 μm from the soma which justifies their wide-field morphology.

2. Need to be more specific when results only apply to vertical or horizontal orientation selective neurons. The grouping of results strongly suggests the model generalizes to

both horizontal and vertical orientation selective neurons. The evidences presented, although suggestive, does not adequately support this generalization.

Horizontal and vertical OS RGCs are plotted separately for their light step responses (Supplementary Fig. 2), their morphology (Fig. 2), their spatial and temporal tuning (Supplementary Fig. 3), and their flashed bar responses (Supplementary Fig. 4). Green and orange points represent data from individual vertical and horizontal OFF OS RGCs, respectively, in Figs. 4,5,6,8, and Supplementary Fig. 9. We mention in the text wherever we found significant differences between OFF vOS and OFF hOS RGCs and that data are pooled when we did not find significant differences. We would be happy to add additional clarifications if there are specific places where we failed to make this distinction.

3. In the analysis of input currents in Figure 4e it is unclear whether you are analyzing the early or late phase of the response or what spot sizes were presented.

The input currents were analyzed in a 0-200ms temporal window at E_{cat} and 500-1000ms temporal window at E_{Cl} . These details have been added in the figure legend. Spot sizes are mentioned on the x-axis of Fig. 4e.

4. Supplementary Figure 3. A-c and b-d do not appear to be the same cells. The illustration of bar angle in a does not match x-axis.

These issues have been fixed in Supplementary Fig. 4.

References

1. Veruki, M. L. & Hartveit, E. Meclofenamic acid blocks electrical synapses of retinal All amacrine and on-cone bipolar cells. *Journal of Neurophysiology* **101**, 2339–2347 (2009).
2. Srinivas, M., Hopperstad, M. G. & Spray, D. C. Quinine blocks specific gap junction channel subtypes. *Proceedings of the National Academy of Sciences* **98**, 10942–10947 (2001).

REVIEWERS' COMMENTS:

Reviewer #1 (Remarks to the Author):

The authors addressed all of my concerns.

Reviewer #2 (Remarks to the Author):

The authors have responded to my last few comments, and in my opinion the manuscript is now acceptable for publication.

In the last submitted version, most of Fig. 9 is not appearing, and I assuming it is the same as the last version.

Reviewer #3 (Remarks to the Author):

I am satisfied with the authors answers.

Minor Comments.

1. The diagram of the stimuli in Supplementary Figure 4a remains incorrect.